# PredGen: Accelerated Inference of Large Language Models through Input-Time Speculation for Real-Time Speech Interaction

**Shufan Li , Aditya Grover**
University of California, Los Angeles

## Abstract

Large Language Models (LLMs) are widely used in real-time voice chat applications, typically in combination with text-to-speech (TTS) systems to generate audio responses. However, their large size often leads to noticeable latency between the end of user input and the start of audio output, resulting in suboptimal user experiences. This latency is particularly evident when LLMs are deployed as single-user voice assistants on consumer-grade hardware with limited computing capacity. We discovered that this latency is primarily dominated by the time it takes for the LLMs to generate the first sentence, which is required as input by the TTS systems that synthesize audio responses on a sentence-by-sentence basis. To address this bottleneck, we propose Predictive Generation (PredGen), a novel framework that mitigates—or even eliminates—this delay through speculative decoding at input time. PredGen generates candidate responses while the user is still speaking, enabling the system to begin TTS processing with minimal delay. Simulated experiments on the Lmsys and MT-Bench datasets show that the proposed method can effectively reduce the latency by around 2× across a wide range of use cases, while incurring only minimal additional computation cost at input time—computation that would otherwise go unused. Code is available at https://github.com/jacklishufan/PredGen

## 1 Introduction

Large Language Models (LLMs) have made considerable progress in recent years in their ability to address a wide range of user queries (Achiam et al., 2023; Grattafiori et al., 2024; Touvron et al., 2023; AI, 2023; Yang et al., 2024). These capabilities make them ideal choices for implementing real-time voice assistants, which can conduct seamless voice conversations with human users.

There are two common strategies for implementing these voice assistants. The first strategy is to use a cascade system that combines an LLM with an automatic speech recognition (ASR) model and a text-to-speech (TTS) model (Huang et al., 2024). To optimize latency, such systems typically invoke the TTS model on a sentence-by-sentence basis instead of waiting for the entire LLM output. Their overall latency consists of the time it takes to generate the first sentence in text, or time-to-first-sentence (TTFS), and the latency incurred by the TTS system.

The second approach involves training an integrated speech-language model that directly generates audio outputs (Zhang et al., 2023; 2024; Fang et al., 2024; Xie & Wu, 2024; Mitsui et al., 2024). These systems typically offer lower latency, as they can generate audio outputs before finishing the first sentence. However, such benefits come with trade-offs. First, they consume substantial compute and voice data during training. Second, state-of-the-art models in this category often lag behind state-of-the-art language models. Lastly, they lack the flexibility of the cascade system. For example, a cascade system can easily incorporate a newly released LLM for better performance or adopt customized LLMs and TTS systems tailored to specific tasks. In contrast, adapting integrated speech-language models requires

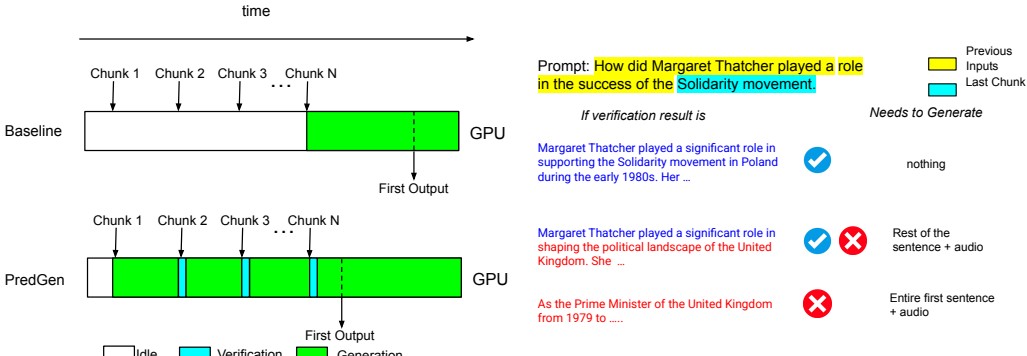

(a) PredGen performs iterative speculative generation and verification while the user is still speaking to reduce latency.

(b) Illustration of the Verification Process. There are three possible outcomes: Accept (Top), Partial Accept (Middle), and Reject (Bottom).

Figure 1: The overall pipeline of PredGen.

training the entire system end-to-end on audio-text data, which is expensive and not always feasible.

In this work, we aim to optimize the latency of the cascade system while preserving its flexibility. In particular, we focus on local deployment on consumer-grade hardware. Compared with alternatives that use external APIs, local deployment offers unparalleled advantages in data security, privacy, and cost.

Formally, we make the following assumptions in designing PredGen. First, the entire system is deployed locally on the user's machine. Second, the user has access to consumer-grade GPUs. Third, since the system is deployed as a standalone application, it will have a single user with a batch size of 1 during inference. Lastly, the user may be willing to fine-tune a model using parameter-efficient techniques such as LoRA (Hu et al., 2022) with a reasonable budget (less than $1000). However, an ideal system would not require fine-tuning.

Our setup leads to several unique characteristics compared with previous works (Shikhar et al., 2025; Chen et al., 2024a) that focus on server-side deployment with top-tier GPUs. We observe that given a complete input sentence, TTS systems such as Zonos (Zyphra, 2025) exhibit similar latency—around 200ms—on both an H100 and an RTX 3090, as TTS models are relatively small. However, for a 7B-scale LLM, the H100 offers significantly higher throughput (approximately 4×) compared to the RTX 3090. This means that, in our setup, latency is dominated by the time it takes for the LLM to generate the first complete sentence. Additionally, while server-side multi-user applications can achieve high utilization through techniques such as continuous batching, in our setup, the GPU remains idle during user input—creating an opportunity to use this underutilized compute for acceleration.

Based on these assumptions, we develop PredGen, an acceleration framework for cascade voice-chat systems. Upon receiving the first chunk of user input, PredGen first generates a candidate response by guessing the user's intention. We also preemptively invoke the TTS system to synthesize the first sentence. Upon receiving subsequent chunks of user input, we iteratively update our guesses through interleaved verification and speculative generation. Finally, upon receiving the complete user input, we perform a final verification. There are three possible outcomes. In the best-case scenario, the first sentence in the candidate response is accepted, and we can directly output the pre-synthesized audio without delay. In less optimal cases, if only part of the first sentence is accepted, we still reduce latency by skipping the accepted tokens. In the worst-case scenario, if no tokens in the first sentence are accepted, we incur a small overhead compared to a no-optimization baseline due to the extra verification process. The overall framework is illustrated in Figure 1.

One of the challenges is that the LLM can become confused with partial prompts, as they are not commonly seen in pretraining data. We address this by incorporating a system prompt

that instructs the LLM to guess the user's intention based on partial inputs. Additionally, we incorporate an optional fine-tuning strategy that further improves the LLM's ability to predict appropriate responses based on incomplete inputs.

We conducted extensive experiments on a variety of datasets, including MT-Bench, LMSys, GSM8K, and MMLU-Pro. Empirical results show that PredGen offers an average of $2\times$ reduction in latency and can almost eliminate latency in some cases.

## 2 Related Works

### 2.1 LLM-Powered Real-time Voice Chat

LLM-powered real-time voice chat applications have drawn considerable interest. The earliest works build cascade systems that combine ASR and TTS with an LLM. AudioGPT (Huang et al., 2024) combines an LLM with Whisper (Radford et al., 2023) for speech recognition and FastSpeech2 (Ren et al., 2020) for speech synthesis. Recent works focus on developing integrated speech-language models. SpeechGPT (Zhang et al., 2023) uses a speech tokenizer (Hsu et al., 2021) to convert speech signals into discrete tokens and builds a language model on these tokens. A vocoder (Kong et al., 2020) is used to convert the generated tokens back into audio signals. Mini-Omni (Xie & Wu, 2024) uses continuous latent features from Whisper to represent speech signals and incorporates a multi-layer parallel decoding mechanism (Copet et al., 2023) for low-latency streaming. IntrinsicVoice (Zhang et al., 2024) introduces GroupFormer, which reduces the length of audio token sequences to improve latency. Other works in this category (Chen et al., 2024c;b; Nguyen et al., 2025; Du et al., 2023) generally follow a similar framework that combines a speech tokenizer, LLM, and a vocoder, with different design choices for specific components. While these approaches offer advantages over cascade systems—such as lower latency, reduced accumulation error, and the ability to incorporate paralinguistic features (e.g., tone, pitch)—they are less flexible and require substantial compute and data during training. It is difficult for an average user to integrate a newly released foundational LLM for better performance or adopt customized LLMs and TTS systems tailored to specific tasks. We offer further discussion on flexibility in Appendix C.2.

### 2.2 Speculative Decoding

The canonical speculative decoding (SD) approach (Chen et al., 2024d; Leviathan et al., 2023) uses a fast draft model to generate draft tokens and employs the base model to verify them in parallel. It recovers the token distribution of the base model while offering considerable speedups. To reduce the overhead of the draft model, recent designs (Cai et al., 2024; Li et al., 2024a;b) employ a self-speculation technique that directly adds additional heads to the base model to generate draft tokens. Several works also explored Jacobi Decoding (Santilli et al., 2023), such as Lookahead (Fu et al., 2024) and CLLM (Kou et al., 2024). These works treat autoregressive generation as a fixed-point iteration process and decode multiple tokens in parallel until convergence. In the standard SD paradigm, the complete user input is always available. SD aims to improve the average generation speed, measured in throughput (tokens/s), after receiving the full user prompt. By contrast, PredGen investigates scenarios involving streaming user input and focuses on the latency to the first chunk of audio output.

### 2.3 Accelerated Inference with Streaming User Inputs

Few previous works have explored scenarios involving streaming user inputs. Livemind (Chen et al., 2024a) uses a 7B model to perform intermediate reasoning steps while the user is still typing. These intermediate results are then used to accelerate inference of a 70B model on an A100 server after the user finishes their input. It focuses on accelerating large model deployment on the server side with top-tie GPUs. By contrast, PredGen targets single-user local deployment with voice outputs using smaller models. We argue that PredGen's setup is more realistic, as it is rare for server-side applications to serve a single user at any given time.

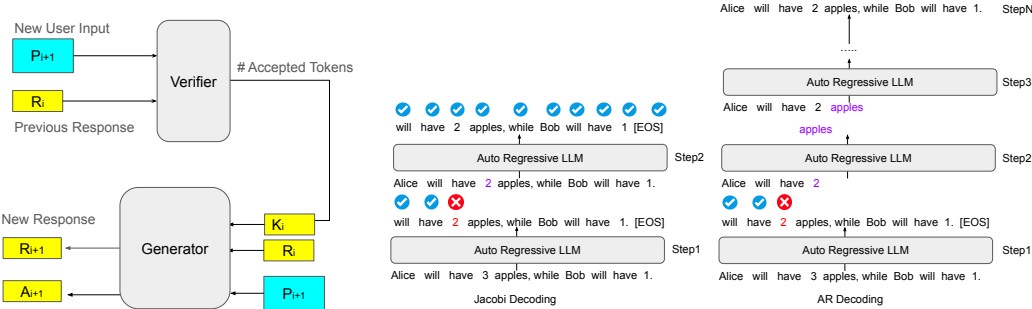

(a) PredGen's Inference Loop.  (b) Illustration of Jacobi Decoding

Figure 2: Algorithm design of PredGen: (a) Upon receiving a partial prompt $P_{i+1}$, we use a verifier to accept $k_i$ tokens from $R_i$. We then create an updated response $R_{i+1}$ and synthesize the audio of its first sentence $A_{i+1}$. (b) We illustrate the Jacobi decoding process and compare it with AR decoding. Jacobi decoding is more efficient in this particular example.

## 3 Methods

We model real-time user inputs as a stream of partial prompts $P_1, P_2, \dots$, where $P_i$ becomes available only after time $t_i$. Suppose the full prompt is "Picture yourself as a 100-year-old tree...," then we could have $P_1$ = "Picture," $P_2$ = "Picture yourself as" and so forth. We assume an input rate of 120 words per minute, or 600 characters per minute, which is the average for conversational speech (Barnard, 2018). After receiving the initial input $P_1$, the model generates a candidate text response $R_1$ and invokes the TTS model to synthesize the first sentence $A_1$. The model then polls the next user input $P_2$ at the current timestamp $t_2$.

Upon receiving a new user prompt $P_{i+1}$, we first employ a verifier to decide how many tokens from the beginning of $R_i$ can be accepted. This number is $k_i$. In the three examples in Figure 1b, $k_i$ is the number of tokens in the accepted portion of the answer (highlighted in blue). We then invoke a subroutine `predictive_generate` to create an updated text response $R_{i+1}$ and synthesize the audio of its first sentence $A_{i+1}$ using $k_i$, $P_{i+1}$, and $R_i$. This process repeats until all user inputs are consumed at iteration $N$, at which point we perform a final verification using prompt $P_N$ and the previously generated text response $R_{N-1}$. If the first sentence is fully accepted, we directly output the previously synthesized audio $A_{N-1}$. Otherwise, we invoke `predictive_generate` again to produce the final audio output. Since in most cases, at least part of the first sentence will be accepted, this approach reduces overall latency by generating fewer tokens to complete the first sentence. After obtaining the audio for the first sentence, we continue to generate the rest of the response and stream the audio output each time a sentence is generated. Because LLM generation speed is significantly faster than the audio play speed, the remainder of the output forms a smooth audio stream. This process is documented in Algorithm 1 and illustrated in Figure 2a.

### 3.1 Design of Verifiers

The verifier takes in a partial prompt $P$ and a previously generated candidate response $R$, and computes the number of accepted tokens $K$. We experimented with three types of verifiers: Greedy, Top-K, and Reflection.

#### 3.1.1 Greedy/Top-K Verification

Greedy verification is the simplest variant of the verification process. Given a prompt $P$ and candidate response $R$, we first concatenate the two into a single input sequence $S = \text{Concat}(P, R)$ of length $L$. We then perform a forward pass using the language model and obtain the log likelihood for the next-token prediction, $Q = \text{LLM}(S) \in \mathbb{R}^{L \times |V|}$, where $|V|$ is the vocabulary size. Finally, we check whether the ArgMax predictions of $Q$ are

---

**Algorithm 1** PredGen's Inference Pipeline

---

**Require:** A stream of partial user prompts $P_1, P_2, ...$ (e.g., at a rate of 600 characters per minute)
 1: **// Initialization: Process the first prompt**
 2: Receive $P_1$
 3: $k_0 \leftarrow 0$
 4: $R_0 \leftarrow ""$
 5: $(R_1, A_1) \leftarrow \texttt{predictive\_generate}(k_0, P_1, R_0)$
 6: $i \leftarrow 1$
 7: **while** User Input Is Not Done **do**
 8:     **// Poll the next user input**
 9:     $t_{i+1} \leftarrow \texttt{now}()$
10:     At time $t_{i+1}$, receive new prompt $P_{i+1}$
11:     **// Verification: Determine accepted part of previous response**
12:     $k_i \leftarrow \texttt{verifier}(P_{i+1}, R_i)$
13:     **// Predictive update: Generate updated response**
14:     $(R_{i+1}, A_{i+1}) \leftarrow \texttt{predictive\_generate}(k_i, P_{i+1}, R_i)$
15:     $i \leftarrow i+1$
16: **end while**
17: **// Final Verification and Output**
18: $N \leftarrow i$
19: Use final prompt $P_N$ and $R_{N-1}$ to perform a final verification of the first sentence.
20: **if** the first sentence is fully accepted **then**
21:     yield $A_{N-1}$
22: **else**
23:     $(R_N, A_N) \leftarrow \texttt{predictive\_generate}(k_{N-1}, P_N, R_{N-1})$
24:     yield $A_N$
25: **end if**
26: **// Continue generating the remainder of the response**
27: Generate and stream the remaining text and corresponding audio output continuously.

---

consistent with the tokens in $S$, and accept the first $K$ tokens that are consistent. Top-K verification is a relaxed version of greedy verification. Instead of requiring an exact match, we accept all tokens in $S$ that are among the top $K$ most likely tokens.

When performing greedy or top-K verification, we also populate the KV cache with the prompt $P$ and the accepted portion of the candidate response $R$, since we resume generation from the first rejected token.

### 3.1.2 Self-Verification via Reflection

Several studies have shown that LLMs can serve as critics to evaluate the quality of text responses (Gu et al., 2024; Tan et al., 2024). Given a prompt $P$ and candidate response $R$, we formulate them into a judge prompt that asks the LLM to evaluate whether the partial prompt is consistent with the first sentence of the candidate answer. We then compute the log-likelihood of the LLM generating a single-word answer—"yes" or "no"—through a forward pass. If "yes" has a higher probability, we accept the entire first sentence; otherwise, we fall back to greedy verification.

This strategy offers the unique advantage of accepting or rejecting an entire segment based on the semantics of the prompt and candidate response, rather than relying on token-level probabilities. For example, the question "Tell me about your favorite country in Asia" has many valid answers. If the candidate response is "My favorite is [X] because ...", [X] may be a plausible answer that is nonetheless rejected by greedy or top-K verification. In contrast, the self-reflection process can recognize that the response is consistent and accept it.

Unfortunately, because the input used for verification is not a simple concatenation of $P$ and $R$, we cannot prefill the KV cache during the verification process.

### 3.2 Design of Predictive Generation

We consider two implementations for the subroutine `predictive_generate`$(k, P, R)$: standard autoregressive (AR) generation and Jacobi decoding with Consistent Large Language Models (CLLM).

### 3.3 AR Generation

Given the number of accepted tokens $k$, the current partial prompt $P$, and a candidate response $R$ (generated based on a previous partial prompt), we first concatenate the prompt $P$ and the accepted tokens $R[0 : k]$ into a single sequence $S$. We then perform standard autoregressive decoding to complete $S$.

AR generation is simple and flexible, as it makes no assumptions about the LLM being used and does not require fine-tuning the underlying model. It also works with any verifier. However, since the prompt $P$ may be incomplete, the LLM might get confused and refuse to answer, or respond with a question asking for clarification. To mitigate this, we incorporate a system prompt. Further details are provided in Appendix A.3.

### 3.4 Jacobi Decoding with CLLM

Jacobi decoding accelerates LLM inference based on the notion that greedy autoregressive decoding induces a fixed-point integration in token space. Specifically, if a sequence $S$ is generated by an LLM using greedy autoregressive decoding, then performing an additional forward pass on $S$ and obtaining the next-token prediction will yield the same sequence shifted by one token.

Hence, we can construct a fixed-point iteration by initializing the sequence $S_0 = \text{Concat}(P, R)$, where $P$ is the current partial prompt and $R$ is the candidate response generated from a previous prompt. We then perform updates

$$S_{i+1} = \text{Concat}(S_i[0], \text{ArgMax}(\text{LLM}(S_i))[0 : L - 1]) \tag{1}$$

where $L$ is the length of $S_i$, until $S_i$ converges.

We provide an illustrative example in Figure 2b. Suppose the candidate $R$ is "Alice will have 3 apples, while Bob will have 1.", and the greedy target is "Alice will have 2 apples, while Bob will have 1." Jacobi decoding corrects the sequence in two steps. By contrast, standard AR generation would first backtrack to the accepted substring "Alice will have", then generate the remaining words ["2", "apples", ...] one by one.

While Jacobi decoding offers considerable theoretical advantages, prior work shows that directly applying it to a base AR model may increase latency, as the base model is not trained to effectively correct itself through Jacobi iterations. Specialized training is required to adapt a standard AR model for Jacobi decoding. We follow the CLLM technique (Kou et al., 2024) to adapt the base model using a calibration dataset. The compute budget is kept under $1000. Further details on CLLM training and compute budget are provided in Appendix A.4 and Appendix C.1.

### 3.5 Text-to-Speech

We incorporate Zonos (Zyphra, 2025), a 1.6B Mamba-Transformer hybrid model, as our TTS system due to its state-of-the-art performance among open-source models. It takes a prefix condition consisting of text tokens and a reference audio sample with the desired voice characteristics, then generates audio outputs autoregressively.

When generating $A_i$, we produce only the fixed number of audio chunks required to resume streaming and save the intermediate state of the TTS model. When $A_i$ is eventually used (Line 21 of Algorithm 1), we first play the regenerated audio chunks while resuming autoregressive audio generation from the saved state. Further details are provided in Appendix A.5.

## 4 Experiments

### 4.1 Setup

We conducted real-time simulations of user inputs on four datasets: Lmsys (Zheng et al., 2023a), MT-Bench (Zheng et al., 2023b), GSM8K (Cobbe et al., 2021), and MMLU-Pro (Wang et al., 2024). Lmsys does not have a standard validation split, so we randomly sampled 100 prompts as the test set. For MT-Bench, we used all 80 prompts. GSM8K and MMLU-Pro are significantly larger, so we randomly selected 100 prompts for testing due to the slow speed of real-time simulation.

For Lmsys and MT-Bench, we report scores evaluated by a GPT-4 judge model, following the MT-Bench setup. For GSM8K and MMLU-Pro, we report accuracy. Overall, Lmsys and MT-Bench are more representative of our intended use cases, as they include diverse conversations. GSM8K focuses on math problem-solving, and MMLU-Pro evaluates general knowledge. We include these benchmarks to provide additional evaluation of sample quality, as accuracy is more interpretable than GPT-4 scores.

We do not simulate the automatic speech recognition (ASR) component, as its latency depends on the choice of model and hardware. There are also many proprietary solutions available exclusively on specific platforms (e.g., macOS, Zoom) that achieve similar transcription quality. In this work, we focus on the latency of the LLM and TTS systems, which is orthogonal to ASR improvements. We simulate user input by directly streaming text to the LLM.

We focus on three key metrics: Time-to-First-Sentence (TTFS), Number-of-Inference-to-First-Sentence (NFETFS), and Audio Latency (abbreviated as "Latency"). TTFS measures the time it takes for the LLM to output the first sentence after receiving the final chunk of user input. NFETFS measures the number of LLM forward passes before the first sentence is output. An NFETFS of 1 means the entire first sentence was accepted during the final verification, and the pre-synthesized audio chunks can be directly used. Audio Latency measures the time between receiving the final chunk of user input and the beginning of audio output. This is not simply the sum of TTFS and TTS latency, because in some cases the first sentence is accepted and audio can be played immediately without re-invoking the TTS model. For this reason, we include a TTS system in our simulation rather than omitting it like the ASR component.

For most experiments, we use Qwen2.5-7B-Instruct (Yang et al., 2024) as the base model. We also include results from a range of other models such as LLaMA 3 (Touvron et al., 2023) and Mistral (AI, 2023) to demonstrate the flexibility of PredGen. We use Zonos (Zyphra, 2025) as the TTS system for all experiments. All evaluations were conducted on a single RTX A5000 GPU with 24GB of VRAM, which is comparable in performance to common consumer-grade GPUs in terms of VRAM and FLOPS (NVIDIA, 2025). Its price range is also below that of top-tier GPUs like the RTX 4090 or 5090, making it a plausible candidate for real-world deployment.

### 4.2 Main Results

We report our main results using Qwen2.5-7B-Instruct as the base model. For training-free AR generation, we experimented with the greedy verifier (PredGen-Greedy), top-3 verifier (PredGen-Top-3), and self-reflection verifier (PredGen-Reflection). For Jacobi decoding with CLLM (PredGen-CLLM), we use the greedy verifier, as CLLM currently only supports greedy decoding. As a baseline, we implement a naive cascade system. We adopt the same TTS system for the baseline, with streaming mode also enabled. The latency is measured according to the output of the first audio chunk, not the synthesis of a complete sentence.

We report results across four datasets—Lmsys, MT-Bench, GSM8K, and MMLU-Pro—in Table 1. All variants of PredGen achieve at least a 1.6× speed-up in terms of audio latency. PredGen-Greedy is a lossless acceleration method that replicates the baseline output exactly; it achieves a 1.6× speed-up on MT-Bench and 1.9× on Lmsys. PredGen-Top-3 and PredGen-Reflection achieve greater speed-ups with a small trade-off in sample quality, as measured

| Method | TTFS↓ | NFETFS↓ | Latency↓ | Score↑ | Speedup↑ |
|---|---|---|---|---|---|
| | | *Lmsys* | | | |
| Qwen2.5-7B-Instruct | 488 | 17.9 | 726 | 8.23 | |
| +PredGen-Greedy | 248 | 9.7 | 374 | 8.23 | 1.9× |
| +PredGen-Top-3 | 171 | 6.4 | 263 | 8.23 | **2.8×** |
| +PredGen-Reflection | **224** | **6.2** | **311** | 8.23 | 2.3× |
| +PredGen-CLLM | 295 | 9.8 | 410 | **8.79** | 1.8× |
| | | *MT-Bench* | | | |
| Qwen2.5-7B-Instruct | 992 | 35.0 | 1233 | 8.27 | |
| +PredGen-Greedy | 623 | 24.4 | 764 | 8.27 | 1.6× |
| +PredGen-Top-3 | 511 | 20.0 | 612 | 8.22 | 2.0× |
| +PredGen-Reflection | 405 | 15.0 | **497** | 8.07 | **2.6×** |
| +PredGen-CLLM | **386** | **14.3** | 529 | **8.51** | 2.3× |
| | | *GSM8K* | | | |
| Qwen2.5-7B-Instruct | 1320 | 42.8 | 1559 | **0.83** | |
| +PredGen-Greedy | 1174 | 38.7 | 1371 | **0.83** | 1.1× |
| +PredGen-Top-3 | **544** | 14.7 | **627** | 0.80 | **2.5×** |
| +PredGen-Reflection | 736 | **12.4** | 779 | 0.80 | 2.0× |
| +PredGen-CLLM | 1023 | 29.7 | 1194 | **0.83** | 1.3× |
| | | *MMLU-Pro* | | | |
| Qwen2.5-7B-Instruct | 1421 | 51.6 | 1670 | **0.50** | |
| +PredGen-Greedy | 984 | 35.7 | 1120 | **0.50** | 1.5× |
| +PredGen-Top-3 | **302** | **10.0** | **361** | **0.50** | **4.6×** |
| +PredGen-Reflection | 994 | 40.5 | 1107 | **0.50** | 1.5× |
| +PredGen-CLLM | 665 | 22.7 | 805 | **0.50** | 2.1× |

Table 1: Main Results Across Four Datasets. Latency refers to Audio Latency. Latency and TTFS are reported in ms.

by GPT-4 scores and accuracy. PredGen-CLLM offers comparable sample quality to the baseline and PredGen-Greedy, while achieving lower average latency. We note that average speed-up is lower on GSM8K and MMLU-Pro, possibly due to the complexity of instructions. For example, GSM8K prompts contain many numerical values, which are harder to infer from partial inputs compared to conversational prompts in MT-Bench and Lmsys.

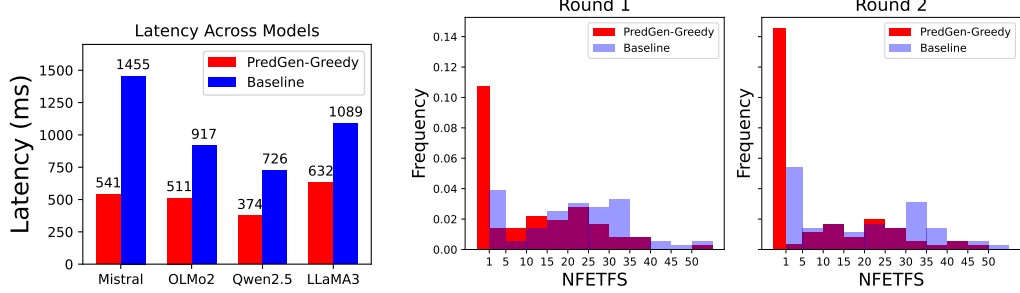

(a) Generalization of PredGen on a variety of base models

(b) Breakdown of Number of Inference Steps on MT-Bench.

Figure 3: Additional Experiments: (a) We report audio latency of four different LLMs around 8B scale on Lmsys dataset. (b) We report the NFETFS on MT-bench for each round of the conversation.

## 4.3 Generalization to Other Base Models

We also study whether PredGen generalizes to base models beyond Qwen2.5. We conducted experiments on the Lmsys dataset with Mistral-7B-Instruct (AI, 2023), OLMo-2-1124-7B-

Instruct (OLMo et al., 2024), and LLaMA3-8B-Instruct (Touvron et al., 2023). Figure 3a reports audio latency by comparing the baseline (with no optimization) to PredGen-Greedy. Since PredGen-Greedy replicates the baseline output exactly, we did not conduct GPT-4 score evaluations due to their cost. Across all base models, PredGen achieves notable speed-ups. Most significantly, using Mistral-7B-Instruct as the base model, PredGen achieves a $2.7\times$ speed-up. Unlike other methods such as (Ren et al., 2020), which require fine-tuning a base LLM for audio generation, our approach highlights the flexibility of a cascade system enhanced by PredGen. It adapts to different base models in a training-free manner.

## 5 Discussions

### 5.1 Feasibility and Effectiveness of PredGen

In this section, we examine the effectiveness of PredGen through the lens of the acceptance rate. In standard speculative decoding (SD), acceptance is computed on individual tokens. In PredGen, the acceptance process is more nuanced, as the verifier can yield three different outcomes (Figure 1b). We visualize the NFETFS metric across two-round conversations from MT-Bench in Figure 3b. When NFETFS equals 1, the system can directly output audio after the verification process. More than 10% of the conversations in both rounds have NFETFS = 1, indicating that audio latency is nearly eliminated for those turns. This percentage is even higher in the second round, presumably because follow-up questions are easier to anticipate, as the model has access to both conversation history and partial input for the current round. Overall, these results demonstrate the feasibility and effectiveness of PredGen for real-time voice chat applications.

### 5.2 Connection with Standard Speculative Decoding

While traditional speculative decoding (SD) methods can achieve comparable speed-ups without input-time speculation, we consider them orthogonal to the focus of this work. Notably, although standard SD can achieve up to $2\times$ speed-up in terms of throughput (tokens per second), only PredGen achieves a best-case latency of 1 NFE—when the entire first sentence is accepted—which is not possible in standard SD. As shown in Figure 3b, up to 14% of inputs in MT-Bench allow us to directly output pre-synthesized audio after a single verification step. In contrast, standard SD may incur additional startup overhead. In fact, it is possible to integrate regular SD methods within PredGen, using them as a faster "base model." Since PredGen reduces the number of tokens that need to be generated by the base model to reach the first sentence, it can further accelerate speculative decoding. We leave this integration for future work.

## 6 Conclusion

In this work, we propose PredGen, a flexible acceleration framework for cascade, LLM-powered, real-time voice chat applications. Our approach introduces a novel direction that utilizes under-exploited GPU compute during user input time to reduce system audio latency. We explored various design choices and proposed multiple variants catering to different needs. PredGen-Greedy replicates the exact output of the base model while achieving noticeable speed-ups. PredGen-Top-K and PredGen-Reflection allow users to trade off a small amount of sample quality for greater speed. For users willing to allocate a modest training budget, PredGen-CLLM offers the same sample quality as PredGen-Greedy, with even lower latency. Cascade systems offer unparalleled flexibility compared to integrated alternatives. However, their potential is often overlooked due to high latency, error accumulation, and limited paralinguistic modeling. PredGen addresses the critical limitation of high latency. We hope our work inspires future research to tackle the remaining challenges in cascaded systems. We provide further discussions of limitations in the Appendix.

## 7 Acknowledgement

AG would like to acknowledge support from NSF CAREER Grant #2341040, Schmidt Sciences Early Career Fellowship, and Amazon Research Award.

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

# A  Implementation Details

## A.1  Definition of a "Sentence"

In our pipeline, we define a "sentence" as a substring delimited by common sentence-ending punctuation such as ".", "?", and "!". We do not consider sub-sentences delimited by symbols such as ",", ":" because they often lack sufficient context for the TTS system to generate coherent audio with appropriate tone and pacing.

## A.2  Self-Reflection Verification

When performing self-verification to determine whether an entire sentence should be accepted, we use the prompt shown in Table 2.

| Self-Reflection Prompts |
|---|
| `<|im_start|>user`
`You are given an incomplete prompt and the model's speculative partial`
`answer.`
`Please judge whether the partial prompt is consistent with the model's`
`answer.`
`Partial Prompt: {{partial_prompt}}`
`Partial Answer: {{partial_answer}}`
`<|im_end|>`
`<|im_start|>assistant`
`{{yes/no}}` |

Table 2: Self-Reflection Prompts

## A.3  AR Generation

Since the base model may become confused when presented with an incomplete prompt, we incorporate a system prompt to discourage undesired responses (e.g., asking the user for clarification). The prompt is shown in Table 3.

| System Prompts for AR Generation |
|---|
| `<|im_start|>system`
`The instruction provided by the user may be truncated. In such cases,`
`respond based on your best guess of the incomplete instruction. Do not`
`complain about incomplete prompts.`
`<|im_end|>` |

Table 3: System Prompt used in AR Generation

## A.4  CLLM Training

### A.4.1  Dataset

We fine-tune Qwen2.5-7B-Instruct (Yang et al., 2024) following the setup of CLLM (Kou et al., 2024) on the Lmsys dataset (Zheng et al., 2023a). We randomly subsample 48k conversations as training examples, ensuring no overlap with the 100 test prompts.

During training, a portion of the prompt is randomly truncated. If a prompt has length $L$, we use the first $k$ tokens, where $k$ is sampled uniformly from $[0, L]$. Since Lmsys is a multi-turn dataset, we randomly select the $i$-th round as the "last round" and discard subsequent rounds $i + 1, i + 2, \ldots$. The assistant response in round $i$ is used as the training target, and we truncate its corresponding user query as described. All conversation history from rounds 0 to $i - 1$ is preserved.

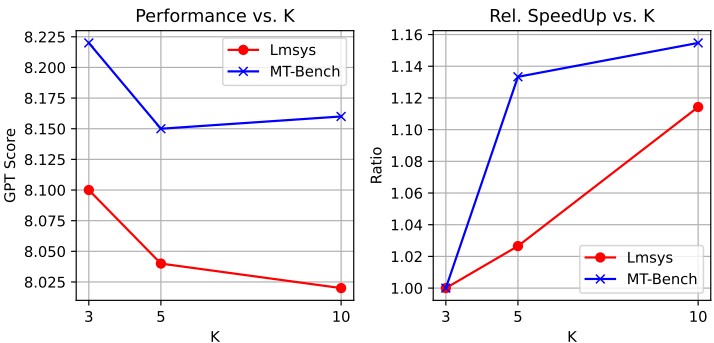

Figure 4: Effect of Top-K Acceptance

### A.4.2 Optimization

We train the model for 6k steps with a learning rate of $1 \times 10^{-5}$ and a global batch size of 8 across 4 A6000 GPUs. We apply LoRA (Hu et al., 2022) and DeepSpeed ZeRO-3 Offload (Rasley et al., 2020) to reduce compute requirements. Training takes approximately 5 days.

### A.5 Text-to-Speech Model

We adopt Zonos (Zyphra, 2025) as our TTS model. To ensure seamless streaming, we generate a 17-chunk audio buffer in the `predictive_generate` subroutine, based on recommendations from the project's GitHub issues.

## B Additional Experiments

### B.1 Trading Latency for Sample Quality

Using the Top-K verifier allows control over the trade-off between speed and sample quality via the hyperparameter *K*. Figure 4 illustrates this effect on Lmsys and MT-Bench. Increasing *K* yields further speed-ups (10–16%) at the cost of a slight reduction in sample quality.

### B.2 Noisy ASR Inputs

We conduct additional experiments using real ASR outputs. We report the results for both clean text inputs and noisy ASR inputs below. To obtain the noisy ASR inputs, we use the Whisper-Live (Radford et al., 2022) library to transcribe synthetic audio created using Chatterbox TTS (Resemble AI, 2025). Notably, Whisper-Live simulates a real-time audio stream by progressively sending audio inputs in chunks of 2048 bytes to the ASR backend. Hence, its output reflects real use cases where the text exhibits a "self-correction" pattern. For example, a list of partial prompts (described in Sec. 3) can be:

```
[
"what is",
"what is one...",
"what is one plus",
"what is one plus two...",
"what is one plus twenty...",
"what is 1 + 23."
]
```

We report the latency and score on the LMSys dataset, as well as the relative speedup. We note that introducing noisy inputs reduces the sample quality score of all methods. However, this is an inherent limitation of ASR systems, which affects the baseline as well.

Compared with respective baselines, our proposed `PredGen` achieves considerable speedup in both cases while maintaining sample quality.

| Method | Latency (ms) | Score | Speedup |
|---|---|---|---|
| Baseline (Clean) | 726 | 8.23 | – |
| Greedy (Clean) | 374 | 8.23 | $1.94\times$ |
| Top-3 (Clean) | 263 | 8.23 | $2.76\times$ |
| Baseline (Noised) | 762 | 8.04 | – |
| Greedy (Noised) | 401 | 8.04 | $1.90\times$ |
| Top-3 (Noised) | 335 | 8.02 | $2.27\times$ |

Table 4: Latency, quality score, and relative speedup of PredGen under clean and noisy ASR inputs.

## C  Additional Discussions

### C.1  Budget Analysis for Fine-Tuning

We train the model using on-premise clusters. Based on current cloud pricing, a $4\times$A6000 node costs approximately \$3.20/hr. Thus, training costs roughly \$384. Factoring in overhead such as data transfer, the total compute cost is estimated to be under \$500.

In contrast, adapting a customized LLM into an integrated system (e.g., Mini-Omni (Xie & Wu, 2024) or Intrinsic Voice (Zhang et al., 2024)) typically requires much more expensive setups. For instance, Mini-Omni was trained on $8\times$A100s for 40,000 steps with a batch size of 192. These nodes cost \$14.32/hr, over $4\times$ the cost of our setup. Even conservative estimates place the training cost above \$10,000.[1]

### C.2  Flexibility in Real-World Use Cases

Integrated speech-language models have clear advantages over cascaded systems: fewer accumulated errors, lower latency, and support for paralinguistic features. However, we argue that there are still many scenarios in which users would prefer a customizable cascade system.

The first use case is customization. Training a speech-language model is prohibitively expensive, but training a custom TTS model (e.g., using VITS (Kim et al., 2021)) is feasible on a single consumer GPU. A cascade system can easily integrate such models. Similarly, users may have custom LLMs (e.g., for roleplay or document QA) or RAG systems that can be swapped into PredGen with no additional training or minimal fine-tuning.

The second use case is general performance. Open-source speech-language models often lag behind state-of-the-art LLMs in performance. Users who choose open-source models for privacy or security reasons cannot benefit from regular updates (e.g., GPT-4o improvements). In contrast, cascade systems allow users to immediately benefit from LLM advancements—whether in accuracy, reasoning, or alignment—without requiring integrated training.

Overall, cascade systems remain a highly flexible and practical option, and PredGen substantially reduces their main bottleneck: latency.

---

[1]Pricing sourced from Lambda Labs at time of writing. Authors have no affiliation with Lambda Labs. This disclosure does not violate double-blind review policy.

## D   Limitations

While PredGen provides significant speed-ups for regular conversations, its benefits are less pronounced for complex tasks like math problem-solving. Future work may explore how to reduce latency for these use cases.

Additionally, our experiments focus on a single-user setting with batch size 1. In multi-user scenarios, PredGen may still be applicable if the number of users is small enough for GPU idle time to permit speculative computation. For instance, a household may share a NAS-based voice assistant. Applying PredGen to such use cases requires more advanced scheduling and batching strategies, which we leave for future work.

## E   Reproducibility Statement

We will release our training and inference code, as well as the subsets of GSM8K, Lmsys, and MMLU-Pro used in our experiments.

