# OpenReview forum: "PredGen: Accelerated Inference of Large Language Models through Input-Time Speculation for Real-Time Speech Interaction"
_colmweb.org/COLM/2025/Conference — COLM 2025_

### Official Review · Reviewer_evVf · 2025-04-24

**Rating:** 6
**Confidence:** 4
**Ethics Flag:** 1

**Summary:**

The paper presents PredGen, a framework aimed at accelerating cascaded ASR-LLM-TTS pipelines. It tackles the inefficiency arising from idle GPU compute time during live user interaction by employing speculative decoding. A key challenge addressed is the handling of partial prompts generated by streaming ASR. PredGen introduces an iterative inference approach to manage this, integrating a verifier for response validation and a predictive generation algorithm. This algorithm reuses verified outputs and generates new responses from updated prompts. A notable aspect is the method's low training overhead (zero or minor training cost). Experiments reportedly demonstrate substantial speedups while maintaining output quality.

**Questions To Authors:**

Further clarification is requested regarding Equation (1) presented in Section 3.4. The current formulation raises ambiguity, as it seems to imply potential modification or editing of the user prompt P. Additionally, the precise relationship between this equation and the number of accepted speculative tokens, k (derived from the verification step), is unclear. Elucidating how k influences the state transition described by Equation (1) would strengthen the reader's understanding of the iterative inference process.

**Reasons To Accept:**

1.	PredGen yields significant speedups without performance loss.
2.	The approach exhibits notable flexibility, requiring minimal to no training overhead for implementation.
3.	The paper effectively articulates a clear and intuitive motivation, and the overall exposition is well-structured and accessible.

**Reasons To Reject:**

1.	The framework's focus appears concentrated on accelerating the LLM component, potentially leaving TTS optimization less explored. A detailed latency breakdown for each system component would help substantiate the primary source of the observed speedups.
2.	A potential significant concern arises from the experimental methodology, which simulates streaming prompts via text segmentation rather than utilizing actual outputs from a streaming ASR system. This simplification may not fully capture real-world complexities, such as the impact of limited context on ASR accuracy in partial inputs, or the effect of ASR self-corrections that can lead to inconsistent intermediate transcripts (i.e, P_{i-1} is not necessarily a prefix of P_{i}). These factors could influence the efficacy of the proposed history response reuse mechanism. Consequently, the reported speedups might represent an optimistic estimate, warranting further validation through experiments incorporating an integrated ASR component.
3.	Certain experimental details require further specification. For example, 1) the strategy employed for simulating the streaming text input needs clarification. 2) the specific ASR system utilized to transcribe the synthesized output audio for calculating evaluation metrics should be explicitly identified.

---

> ### Author Response · Authors · 2025-06-02
>
> **W1. A detailed latency breakdown for each system component would help substantiate the primary source of the observed speedups.**
>
> We already provide two metrics: **Time-to-First-Sentence (TTFS)** and **Latency** in Table 1. TTFS corresponds to LLM latency, and Latency is the total latency (LLM + TTS). Thus, **TTS latency = Latency - TTFS**. The detailed breakdown is as follows:
>
> |              | Latency↓ (ms) | LLM Latency (TTFS)↓ | TTS Latency↓ |
> |--------------|---------------|-----------------------|----------------|
> | Baseline     | 726           | 488                   | 238            |
> | Greedy       | 374 (1.9×)    | 248 (2.0×)            | 126 (2.0×)     |
> | Top-3        | 263 (2.8×)    | 171 (2.9×)            | 92 (2.7×)      |
> | Reflection   | 311 (2.3×)    | 224 (2.2×)            | 87 (2.9×)      |
> | CLLM         | 410 (1.8×)    | 295 (1.7×)            | 115 (2.2×)     |
>
> PredGen reduces latency across all components. Although it does not reduce the time TTS takes to convert text to audio, it avoids running TTS when speculative answers are accepted early.
>
> **W2 A potential significant concern arises from the experimental methodology, which simulates streaming prompts via text segmentation rather than utilizing actual outputs from a streaming ASR system.**
>
> We conduct additional experiments using real ASR outputs. We report the results of clean text input and noisy ASR inputs as below. To obtain the noisy ASR inputs, we use Whisper-Live library to transcribe synthetic audio created using Chatterbox TTS. Notably, Whisper-Live simulate a real-time audio stream by progressively sending audio inputs in chunks of 2048 bytes to the ASR backend, hence, its output reflects real use cases where the text outputs exhibits a "self-correction" pattern. For example, a list of partial prompts $P_1,P_2...$ (described in Sec 3) can be
> [
> "what is",
> "what is one...",
> "what is one plus",
> "what is one plus two...",
> "what is one plus twenty...",
> "what is 1 + 23."
> ]
>
> Note that under this setup no change is required to our Algorithm 1 because it does not explicitly assume $P_{i}$ is a prefix of $P_{i+1}$. We can still perform the verification step by concatenating $P_{i+1}$ with the previously generated answer of $P_{i}$ regardless of their relations. The only detail is that we can no longer reuse the entirety of the KV cache of $P_{i}$. Instead, we reuse the KV cache of the common prefix between $P_{i}$ and $P_{i+1}$.
>
> We report the latency and score on LMsys dataset, as well as the relative speed up, using the clean text input and real ASR stream. We note that introducing noisy inputs reduce the sample quality score of all method. However, this is an inherent limitation of ASR systems, as it affects the baseline as well. Comparing with respective baselines, our proposed PredGen was able to achieve considerable speed up in both cases while maintaining sample quality
>
> |  | Latency (ms) | Score |  Speedup |
> | --- | --- | --- | --- |
> | Baseline (Clean) | 726 | 8.23 | - |
> | Greedy (Clean) | 374 | 8.23 | 1.94x |
> | Top 3 (Clean) | 263 | 8.23 | 2.76x |
> | Baseline (Noised) | 762 | 8.04 | - |
> | Greedy (Noised) | 401 | 8.04 | 1.90x |
> | Top3 (Noised) | 335 | 8.02 | 2.27x |
>
> **W3 Certain experimental details require further specification.**
>
> We clarify that original experiments use character-speed-based segmentation (assuming 600 characters per minute from Sec 3) of the original text prompt from the dataset. This is described in Sec. 4.1 and is a reasonable idealization. Our real-ASR experiments confirm that PredGen remains effective even under real-world transcription behavior (see W2 for detailed setup).
>
> **Q1. Clarification on Equation (1), Section 3.4**
>
> Yes, the variable *k* does not appear in Equation (1) because it only inexplicitly determines how much KV cache we can reuse.
>
> 1. $LLM(S_i)$ is effectively evaluated as $LLM(\text{Cache}, S_i[M:L])$, where $M = |P| + k$. Since *k* tokens are accepted, we can reuse the KV cache up to that point (including the entire prompt and k tokens from the answer). Only the remaining $L-M$ tokens require computation.
>
> 2. Prompt editing is only an illusion: we do not recompute logits or argmax on prompt tokens during the iterative steps because of KV cache. They are effectively assumed to be accepted. That is, $\text{Argmax}(LLM(S_i))[0:M-1] = S_i[1:M]$.
>
> 3. The number of refinement steps in $S_0, S_1, \dots$ depends on the model. For example, correcting `Alice has 3 apples and 2 oranges.` to `Alice has 2 apples and 3 oranges.` could take one or multiple iterations:
>
> ```
> # Possibility 1
> S_0: Alice has 3 apples and 2 oranges.
> S_1: Alice has 2 apples and 3 oranges.
>
> # Possibility 2
> S_0: Alice has 3 apples and 2 oranges.
> S_1: Alice has 2 apples and 2 oranges.
> S_2: Alice has 2 apples and 3 oranges.
>
> ```
>
> While the convergence is guaranteed, its speed depends on the precise model predictions, and is not directly related to k. We will clarify this further in the paper.

---

> > ### Comment · Reviewer_evVf · 2025-06-03
> >
> > Thank you for your clarifications. I also greatly appreciate the additional experiments using real ASR outputs and hope these valuable results can be incorporated into the paper. In light of this, I have a few further points I would like to address:
> > 1. Regarding the prompt editing mechanism, it remains unclear to me whether the assertion that "prompt tokens are always accepted" can be inferred solely from Equation 1. Given that S_0 represents the concatenation of the current prompt (P) and response (R), and this entire sequence S is subsequently edited according to the state transitions defined in Equation 1, this formulation seems to imply that P, as an integral part of S, could also be subject to editing. I would be grateful for a clarification on how prompt immutability is ensured under this scheme.
> > 2. I would like to verify whether the reported benchmark scores are calculated based on the raw predicted text or on the transcribed text from the synthesized speech. I believe incorporating results from the latter approach would offer a more comprehensive evaluation, as it would also reflect the performance of PredGen's optimized TTS component. If such an evaluation based on transcribed speech was conducted, please specify the ASR system that was employed for the transcription.

---

> ### Author Response · Authors · 2025-06-03
>
> **Q1**
>
> We provide a concrete example below.
>
> Suppose the prompt $P$ is `What is the capital of France?`, and the speculated answer $R$ is `The capital of Germany is Berlin`, which was generated when the word `France` was not available.
>
> Before constructing $S_0$ for iteration, we first perform a verification step and accept the tokens `The capital of` ($k=3$). In this step, we already performed an LLM forward call on the string `What is the capital of France? The capital of Germany is Berlin`. (note that the verification string is exactly $S_0$.) We obtained a valid KV cache for the substring `What is the capital of France? The capital of`. Moreover, in the forward call, we can also obtain the next-token predictions corresponding to the words `of Germany is Berlin`. Let's assume they are `France is Berlin [.]`.
>
> Hence, we already obtained $S_1$ after the verification step. $S_1$ is `What is the capital of France? The capital of France is Berlin.` For $S_1$, we can reuse the KV cache of the prefix `What is the capital of France? The capital of`. The actual LLM input consists of only the unaccepted tokens `France is Berlin`, along with the past KV cache, stored as real-valued tensors. In this regard, when computing $LLM(S_1)$, we only get predictions for the unaccepted part `France is Berlin`. Consequently, $\text{ArgMax}(LLM(S_i))$ only changes $S_i$ in the unaccepted part, which is the last 3 tokens in this example.
>
> We recognize that this dynamic is not fully captured in Eq. (1). A clearer formulation might be:
>
> $S_0 = R$
>
> $S_{i+1} = \text{Concat}(S_i[0], \text{ArgMax}(LLM(P, S_i))[0 : L - 1])$
>
> where $LLM(P, S_i)$ only produces next-token predictions for the unaccepted portion of $S_i$.
>
> We note that this formulation still does not fully reflect the complexity of the cache implementation, but it serves as an equivalent illustration of the process. This cached iterative refinement process was originally documented in Appendix C of the CLLM paper [1]. We will incorporate it as background in future versions for clarity.
>
> **Q2**
>
> In the original paper, we directly ran benchmark evaluations on clean text predictions. We incorporate results obtained using Whisper-Live below.
>
> (Note: The baseline and ours-Greedy methods should have the same text outputs, but different TTS outputs due to the non-deterministic nature of our TTS.)
>
> **LMSYS**
> | Method              | Score (Text) | Score (Transcribed) | Δ     |
> |---------------------|--------------|----------------------|--------|
> | Baseline            | 8.23         | 8.20                 | -0.03  |
> | +ours-Greedy        | 8.23         | 8.23                 | 0.00   |
> | +ours-Top-3         | 8.23         | 8.23                 | 0.00   |
> | +ours-Reflection    | 8.23         | 8.21                 | -0.02  |
> | +ours-CLLM          | 8.79         | 8.69                 | -0.10  |
>
> **MT-Bench**
> | Method              | Score (Text) | Score (Transcribed) | Δ     |
> |---------------------|--------------|----------------------|--------|
> | Baseline            | 8.27         | 8.26                 | -0.01  |
> | +ours-Greedy        | 8.27         | 8.23                 | -0.04  |
> | +ours-Top-3         | 8.22         | 8.26                 | +0.04  |
> | +ours-Reflection    | 8.07         | 8.12                 | +0.05  |
> | +ours-CLLM          | 8.51         | 8.48                 | -0.03  |
>
> **GSM8K**
> | Method              | Score (Text) | Score (Transcribed) | Δ     |
> |---------------------|--------------|----------------------|--------|
> | Baseline            | 0.83         | 0.83                 | 0.00   |
> | +ours-Greedy        | 0.83         | 0.83                 | 0.00   |
> | +ours-Top-3         | 0.80         | 0.80                 | 0.00   |
> | +ours-Reflection    | 0.80         | 0.80                 | 0.00   |
> | +ours-CLLM          | 0.83         | 0.83                 | 0.00   |
>
> **MMLU-Pro**
> | Method              | Score (Text) | Score (Transcribed) | Δ     |
> |---------------------|--------------|----------------------|--------|
> | Baseline            | 0.50         | 0.50                 | 0.00   |
> | +ours-Greedy        | 0.50         | 0.50                 | 0.00   |
> | +ours-Top-3         | 0.50         | 0.50                 | 0.00   |
> | +ours-Reflection    | 0.50         | 0.50                 | 0.00   |
> | +ours-CLLM          | 0.50         | 0.50                 | 0.00   |
>
> [1] Kou, Siqi, et al. *"CLLMs: Consistency Large Language Models."* ICLR 2024.

---

> > ### Comment · Reviewer_evVf · 2025-06-04
> >
> > Thank you for your rebuttal. I have raised my score to 6.

---

### Official Review · Reviewer_2rYs · 2025-04-27

**Rating:** 7
**Confidence:** 3
**Ethics Flag:** 1

**Summary:**

This paper introduces PredGen, a method to accelerate cascaded systems of real-time speech interactions on commercial-grade GPUs. Common cascaded systems tend to have high latency in generating the first response, due to multiple systems involved. The authors integrate techniques inspired by speculative decoding to decrease latency. The evaluation results show that the proposed method can effectively reduce latency and maintain performance on several datasets.

**Questions To Authors:**

1. For Table 1, I am a bit surprised that the scores on MMLU-Pro are exactly the same across four conditions. Could the authors confirm to me that this is indeed the case? Thanks.

**Reasons To Accept:**

This paper does have several merits, as detailed below.
1. While the authors only focus on a single user scenario with a batch size of 1, the perspective on the cost-effectiveness is still highly valuable. The optimization focuses on the commercial-grade hardware, which makes it potentially usable for many users without a high-end GPU.
2. The method is potentially generalizable. The acceleration algorithm reduces latency while mostly maintaining the performance as the baseline model in Table 1. The results in Figure 3 further show that the method is generalizable across base models.

**Reasons To Reject:**

1. ASR is not included. The authors argue that they only focus on the LLM and TTS part, which is independent of the ASR component. I do not fully agree with this statement, though I acknowledge that this is somewhat true. The ASR-generated texts are still not perfect in dealing with noises and accents. If the input texts are noisy, then the time it takes to verify can be longer or the performance can suffer. The noisy input conditions can actually be simulated with noisy texts. I won't say this is a reason to reject, but having the ASR-generated noisy texts in the results can make this paper stronger.

---

> ### Author Response · Authors · 2025-06-02
>
> **W1. ASR not included**
>
> We now conduct additional experiments using real ASR outputs. We report the results for both clean text inputs and noisy ASR inputs below. To obtain the noisy ASR inputs, we use the Whisper-Live library to transcribe synthetic audio created using Chatterbox TTS. Notably, Whisper-Live simulates a real-time audio stream by progressively sending audio inputs in chunks of 2048 bytes to the ASR backend. Hence, its output reflects real use cases where the text exhibits a "self-correction" pattern. For example, a list of partial prompts $P_1, P_2, \dots$ (described in Sec. 3) can be:
>
>
> ```
> [
> "what is",
> "what is one...",
> "what is one plus",
> "what is one plus two...",
> "what is one plus twenty...",
> "what is 1 + 23."
> ]
> ```
> We report the latency and score on the LMSys dataset, as well as the relative speedup. We note that introducing noisy inputs reduces the sample quality score of all methods. However, this is an inherent limitation of ASR systems, which affects the baseline as well. Compared with respective baselines, our proposed PredGen achieves considerable speedup in both cases while maintaining sample quality.
>
> |                       | Latency (ms) | Score | Speedup |
> |-----------------------|--------------|-------|---------|
> | Baseline (Clean)      | 726          | 8.23  | –       |
> | Greedy (Clean)        | 374          | 8.23  | 1.94×   |
> | Top-3 (Clean)         | 263          | 8.23  | 2.76×   |
> | Baseline (Noised)     | 762          | 8.04  | –       |
> | Greedy (Noised)       | 401          | 8.04  | 1.90×   |
> | Top-3 (Noised)        | 335          | 8.02  | 2.27×   |
>
>
>
> ---
>
> **MMLU-Pro numbers are exactly the same**
>
> Yes, the numbers are correct. While the outputs of different methods are not identical strings, MMLU-Pro is a multiple-choice task. Unlike GSM8K, where the answer is a numerical string, the final choice remains unchanged despite differing intermediate reasoning steps.

---

> > ### Comment · Reviewer_2rYs · 2025-06-09
> >
> > Thank you for the new experiments. It has addressed my concern and I will raise your score.

---

> ### Author Response · Authors · 2025-06-09
>
> Dear Reviewer
>
> We appreciate your comments and feedbacks. We hope that our rebuttal has addressed your concerns. If you have any further questions, we are happy to clarify.
>
>
> Authors

---

### Official Review · Reviewer_XMLi · 2025-05-13

**Rating:** 5
**Confidence:** 4
**Ethics Flag:** 1

**Summary:**

The paper proposes Predictive Generation (PredGen), a speculative decoding framework that generates candidate responses while the user is still speaking and allowing TTS to begin generating candidate voice responses to potentially reduce delays. The paper presents simulated experiments on the Lmsys and MT-Bench datasets claiming that the proposed method reduced the latency by around 2× across a wide range of use cases.

**Reasons To Accept:**

The paper proposes an interesting idea with Predictive Generation (PredGen), a speculative decoding framework that generates candidate responses while the user is still speaking and allowing TTS to begin generating candidate voice responses to potentially reduce delays. The paper presents simulated experiments on the Lmsys and MT-Bench datasets claiming that the proposed method reduced the latency by around 2× across a wide range of use cases.

**Reasons To Reject:**

The proposed methodology requires that the solution run on a local user machine to allow free GPU cycles (while the user is speaking). To power the proposed solution, the authors use a RTX A5000 24GB GPU, which costs ~$2000. The authors call this GPU a "common consumer-grade GPU". In my opinion, regular users of voice assistants will not own such expensive hardware. I do not foresee practical application for the proposed methodology and consider it to be of low practical impact.

---

> ### Author Response · Authors · 2025-06-02
>
> **Regular users of voice assistants will not own such expensive hardware (RTX A5000), hence I see limited impact.**
>
> We respectfully disagree.
>
> First, the RTX A5000’s price (USD 2000) is significantly cheaper than the current market price of top-tier consumer-grade GPUs such as the RTX 5090 (USD  4000+) and RTX 4090 (USD  3000+), and offers weaker performance. We believe it is representative of consumer hardware for potential users of AI voice chat applications.
>
> Second, to further demonstrate the effectiveness of PredGen, we report additional results on the MacBook Pro with Apple Silicon, which has a considerable user base. Due to lower performance, we use a lightweight Qwen2.5-3B-Instruct instead of the 7B model. We report results on the LMSys benchmark below:
>
> |        | Latency (ms) | Score | Speedup |
> |--------|--------------|-------|---------|
> | Baseline | 3914       | 7.72  | –       |
> | Greedy   | 2164       | 7.69  | 1.8×    |
> | Top-3    | 2002       | 7.98  | 2.0×    |
>
> Notably, PredGen achieves similar speedup as on the RTX A5000. We argue that PredGen may be even more meaningful on lower-tier hardware, where base models are slower. For instance, it reduces latency from 3.9s to 2.0s—more noticeable from a user perspective than the improvement from 0.7s to 0.3s on A5000.

---

> ### Author Response · Authors · 2025-06-09
>
> Dear Reviewer
>
> We appreciate your comments and feedbacks. We hope that our rebuttal has addressed your concerns. If you have any further questions, we are happy to clarify.
>
> Authors

---

### Official Review · Reviewer_iE4i · 2025-05-13

**Rating:** 7
**Confidence:** 2
**Ethics Flag:** 1

**Summary:**

This paper proposes PredGen, a framework for accelerating real-time LLM-powered voice assistants by performing speculative response generation during user input. It reduces time-to-first-sentence (TTFS) latency by generating and verifying candidate responses while the user is still speaking, enabling early TTS synthesis. The system supports multiple verifier strategies and generation modes, and shows up to 2.8× latency reduction with minimal quality loss across diverse datasets.

**Questions To Authors:**

How well does the self-reflection verifier generalize across different model families or prompt domains?

How is correction or interruption handled in user input — can early speculative audio be invalidated and rolled back?

Have you tested PredGen in noisy ASR scenarios, where the initial prompt may be erroneous?

Could traditional speculative decoding (e.g., Medusa) be combined with PredGen to further reduce latency after full input is received?

**Reasons To Accept:**

- Targets a real bottleneck in local voice assistants — the latency before audio starts — and provides a deployable solution for consumer hardware.

- Introduces input-time speculative decoding, which differs from traditional speculative decoding that only starts after input is complete.

- Covers multiple datasets (MT-Bench, LMSys, GSM8K, MMLU), several LLMs (Qwen, LLaMA3, Mistral), and verifier/generator variants.

- Works in a training-free mode (PredGen-Greedy), and offers further gains with optional lightweight CLLM tuning under $500.

- Unlike end-to-end speech-LMs, PredGen allows plug-and-play of LLMs and TTS models.

**Reasons To Reject:**

- Relies on GPU idle time during user input, which may not generalize to shared/multi-user settings without more complex scheduling.

- Gains are modest on complex reasoning tasks like GSM8K and MMLU, where partial prompt prediction is harder.

- Self-reflection verification is promising but might be brittle; the paper lacks discussion on failure modes or robustness across prompt styles/domains.

---

> ### Author Response · Authors · 2025-06-02
>
> **Q1. How well does the self-reflection verifier generalize across different model families or prompt domains?**
>
> In Table 1, we show that self-reflection generalizes to multiple datasets using Qwen as the base model. However, it is not as effective on the MMLU-Pro dataset as on other datasets, presumably due to the complexity of the problem.
>
> To investigate whether the reflection verifier generalizes to other models, we provide additional results using different base models in the table below:
>
> | Baseline | Speedup (Greedy) | Speedup (Ref.) | Score (Baseline/Greedy) | Score (Ref.) |
> |----------|------------------|----------------|--------------------------|--------------|
> | Mistral  | 2.7×             | 2.7×           | 7.54                     | 7.54         |
> | OLMo     | 1.8×             | 2.0×           | 8.03                     | 7.95         |
> | Qwen2.5  | 1.9×             | 2.3×           | 8.23                     | 8.23         |
> | Llama3   | 1.7×             | 2.0×           | 8.10                     | 8.10         |
>
> We report the speedup of the greedy and reflection verifiers on the LMSys dataset using four different models: Mistral-7B-Instruct, OLMo-2-1124-7B-Instruct, Llama-8B-Instruct, and Qwen2.5-7B-Instruct. We also report the scores of the baseline approach and the reflection verifier. Note that since the greedy verifier is lossless, it achieves the exact same score as the baseline.
>
> These results show that self-verification works best with more recent models with stronger reasoning capabilities, such as Qwen2.5 and Llama3. For earlier models such as OLMo and Mistral, it leads to only limited improvements.
>
> Recall that in Sec. 3.1.2, we mentioned that the reflection verifier falls back to the greedy verifier when the model gives a "no" answer during the reflection process. We find that Mistral, in particular, gives a "no" answer in most cases, leading to an almost identical speedup to the greedy verifier.
>
>
>
> **Q2. How is correction or interruption handled in user input — can early speculative audio be invalidated and rolled back?**
>
> Yes. Most ASR systems employ a Voice Activity Detector (VAD). For example, if a user says "[incomplete prompt A] [interruption], actually, what about [prompt B]", the ASR system treats the part after the interruption as a new input. Hence, our system, which relies on ASR input, considers it a separate query and discards past speculative results. Overall, we believe this is a feature of ASR systems and not the focus of our work.
>
> We also provide discussions on ASR autocorrection in Q3.
>
>
>
> **Q3. Have you tested PredGen in noisy ASR scenarios, where the initial prompt may be erroneous?**
>
> Yes. We report the results for both clean text inputs and noisy ASR inputs below. To obtain the noisy ASR inputs, we use the Whisper-Live library to transcribe synthetic audio created using Chatterbox TTS. Notably, Whisper-Live simulates a real-time audio stream by progressively sending audio inputs in chunks of 2048 bytes to the ASR backend. Hence, its output reflects real use cases where the text exhibits a "self-correction" pattern. For example, a list of partial prompts $P_1, P_2, \dots$ (described in Sec. 3) can be:
>
>
> ```
> [
> "what is",
> "what is one...",
> "what is one plus",
> "what is one plus two...",
> "what is one plus twenty...",
> "what is 1 + 23."
> ]
> ```
> We report the latency and score on the LMSys dataset, as well as the relative speedup. We note that introducing noisy inputs reduces the sample quality score of all methods. However, this is an inherent limitation of ASR systems, which affects the baseline as well. Compared with respective baselines, our proposed PredGen achieves considerable speedup in both cases while maintaining sample quality.
>
> |                       | Latency (ms) | Score | Speedup |
> |-----------------------|--------------|-------|---------|
> | Baseline (Clean)      | 726          | 8.23  | –       |
> | Greedy (Clean)        | 374          | 8.23  | 1.94×   |
> | Top-3 (Clean)         | 263          | 8.23  | 2.76×   |
> | Baseline (Noised)     | 762          | 8.04  | –       |
> | Greedy (Noised)       | 401          | 8.04  | 1.90×   |
> | Top-3 (Noised)        | 335          | 8.02  | 2.27×   |
>
>
>
> **Q4. Could traditional speculative decoding (e.g., Medusa) be combined with PredGen to further reduce latency after full input is received?**
>
> In principle, yes. However, in practice, the throughput (i.e., tokens/sec) of LLMs without speculative decoding is typically higher than the audio playback speed (which also aligns with typical human speaking speed). Hence, while traditional speculative decoding may improve the *end-to-end latency* for text generation, it does not affect *when* the user hears the synthesized sentence. We leave this for future work.

---

> > ### Comment · Reviewer_iE4i · 2025-06-06
> >
> > Thanks for you reply. I will keep my score.

---

### Decision · Program_Chairs · 2025-07-08

**Decision:**

Accept

**Comment:**

The paper focuses on latency in real-time voice applications from waiting for the LLM to generate a sequence to be passed into the text-to-speech system. The authors propose an (iterative) speculative decoding approach which proposes multiple candidate responses while the user is still speaking---allowing the TTS to start more quickly. To "filter" the generated candidate responses the authors experiment with several verification methods: greedy, top-k and self-verification via reflection. The method assumes GPU idle time while the user is speaking, and under this assumption achieves up 2.8× latency reduction with little loss in quality.

The reviewers were overall positive for the paper---with some concerns that were mostly addressed during the rebuttal period. Specifically, the reviewers appreciated the fact that the paper addresses a real pain point, uses a natural strategy that's executed well, and requires minimal-to-no training overhead. There was a concern about real (noisy) ASR not being tested, which was added by the authors during the rebuttal period and satisfied the reviewers. There was also a concern about the reliance on GPU idle time (which may not generalize to shared/multi-user settings) and the cost of the hardware the experiments rely on---but ultimately these concerns were considered to not overshadow the contributions of the paper.